# You Can't Count on Luck:
# Why Decision Transformers and RvS
# Fail in Stochastic Environments

**Keiran Paster**
Department of Computer Science
University of Toronto, Vector Institute
keirp@cs.toronto.edu

**Sheila A. McIlraith & Jimmy Ba**
Department of Computer Science
University of Toronto, Vector Institute
{sheila, jba}@cs.toronto.edu

## Abstract

Recently, methods such as Decision Transformer [1] that reduce reinforcement learning to a prediction task and solve it via supervised learning (RvS) [2] have become popular due to their simplicity, robustness to hyperparameters, and strong overall performance on offline RL tasks. However, simply conditioning a probabilistic model on a desired return and taking the predicted action can fail dramatically in stochastic environments since trajectories that result in a return may have only achieved that return due to luck. In this work, we describe the limitations of RvS approaches in stochastic environments and propose a solution. Rather than simply conditioning on the return of a single trajectory as is standard practice, our proposed method, ESPER, learns to cluster trajectories and conditions on average cluster returns, which are independent from environment stochasticity. Doing so allows ESPER to achieve strong alignment between target return and expected performance in real environments. We demonstrate this in several challenging stochastic offline-RL tasks including the challenging puzzle game 2048, and Connect Four playing against a stochastic opponent. In all tested domains, ESPER achieves *significantly* better alignment between the target return and achieved return than simply conditioning on returns. ESPER also achieves higher maximum performance than even value-based baselines.

## 1 Introduction

Offline reinforcement learning (RL) [3, 1, 4, 5] is a promising approach to train agents without requiring online experience in an environment, which is desirable when online experience is expensive or when offline experience is abundant. A recent trend in offline RL has been to use simple approaches that do RL via Supervised Learning (RvS) (e.g., [1, 2, 6–8]) rather than typical value-based approaches. RvS algorithms such as Decision Transformer [1] train a model to predict an action based on the current state and an outcome such as a desired future return. These agents ask the question *"if I assume the desired outcome will happen, in my experience what action do I typically take next."* These methods are popular due to their simplicity, strong performance on offline benchmark tasks [1, 2], and similarity to large generative models (e.g., [9–12]) that continue to show stronger performance on more tasks when training larger models on more data.

However, as we will show, **methods that condition on outcomes such as return can make incorrect decisions in stochastic environments regardless of scale or the amount of data they are trained on.** This is because implicitly these methods assume that actions that end up achieving a particular goal are optimal for achieving that goal. This assumption is not true in stochastic environments, where it is possible that the actions taken in the trajectory were actually sub-optimal and that the outcome was only achieved due to lucky environment transitions. For example, consider the gambling

36th Conference on Neural Information Processing Systems (NeurIPS 2022).

environment in Figure 1. Though there may be many episodes in which an agent gets a positive return from gambling ($a_0$ or $a_1$), gambling is sub-optimal since it results in a negative return in expectation while $a_2$ always results in a positive return. Since RvS takes all of these trajectories as expert examples of how to achieve the goal, RvS will act sub-optimally.

This limitation of RvS approaches in stochastic environments is well-hidden by the majority of benchmark tasks for offline RL (e.g., [13, 14]), which tend to be deterministic or near-deterministic. Locomotion tasks in MuJoCo [15] and Atari games in the Arcade Learning Environment [16] are two such examples. While deterministic tasks can be solved by replaying promising action sequences, stochastic tasks are significantly less trivial to solve and often more realistic, requiring reactive policies to be learned [17]. Many real-world tasks are stochastic, either inherently or due to partial observability, such as having a conversation, driving a car, or navigating an unknown environment. In their current form, approaches such as Decision Transformer [1] are likely to behave in unexpected ways in such scenarios.

One way to view why RvS doesn't work in the gambling environment is that when conditioning on trajectories that achieve a positive reward, the model doesn't get to see any of the trajectories where the same sequence of actions leads to a negative reward. Due to these unrealistic dynamics, there is no policy that would generate this set of trajectories in the real environment, so it doesn't make sense to treat them as expert trajectories. Our insight is that there are certain functions of the trajectory other than return that, when conditioned on, will better preserve the dynamics of the environment. Our approach, called ESPER, realizes this by *conditioning on outcomes that are fully determined by the actions of the agent and independent of the uncontrollable stochasticity of the environment*. While trajectory return is not such an outcome, we show that the *expected* return of behavior shown in a trajectory is, and how to learn such a value. The contributions of this work are as follows:

- We show that RvS-style agents can only *reliably* **achieve outcomes that are independent from environment stochasticity** and only depend on information that is under the agent's control.

- We propose a method to **learn environment-stochasticity-independent representations of trajectories using adversarial learning**, on top of which we label each trajectory with the average return for trajectories with this representation.

- **We introduce several stochastic offline RL benchmark tasks** and show that while RvS agents consistently underperform on the conditioned returns, our approach (ESPER, short for **e**nvironment-**s**tochasticity-inde**pe**ndent **r**epresentations) **achieves significantly stronger alignment between target return and actual expected performance.** ESPER gets **state-of-the-art performance** on these tasks, solving all tasks with near-maximum performance and surpassing the performance of even strong value-based methods such as CQL [4].

## 2   Approach

### 2.1   Problem Setup

We model the environment as an MDP, defined as the tuple $(S, A, T, R, \gamma)$. $S$ is a set of states; $A$ is a set of actions; the transition probabilities $T : S \times A \times S \to [0, 1]$ defines the probability of the environment transitioning from state $s$ to $s'$ given that the agent acts with action $a$; the reward function $R : S \times A \to \mathbb{R}$ maps a state-action transition to a real number; and $0 \le \gamma \le 1$ is the discount factor, which controls how much an agent should prefer rewards sooner rather than later. The performance a policy, defined by $\pi(a|s)$, is typically measured using the cumulative discounted return $\sum_t \gamma^t r_t$ that the policy achieves in an episode.

Central to our work is algorithms that do reinforcement learning via supervised learning (RvS) such as Decision Transformer [1]. These approaches train a model using supervised learning on a dataset of trajectories to predict $p_\mathcal{D}(a|s, R)$ - the probability of next action conditioned on the current state and on the agent getting a cumulative discounted return $R = \sum_t \gamma^t r_t$. At evaluation time, the model is conditioned on a desired target return and the agent takes the actions predicted by the model.

To understand the problem with RvS in stochastic environments, we build off of a general framework where a goal is presented to a policy, the policy acts in the environment, and the agent is scored on how well the goal was achieved as described in Furuta et al. [18]. Mathematically, the agent's

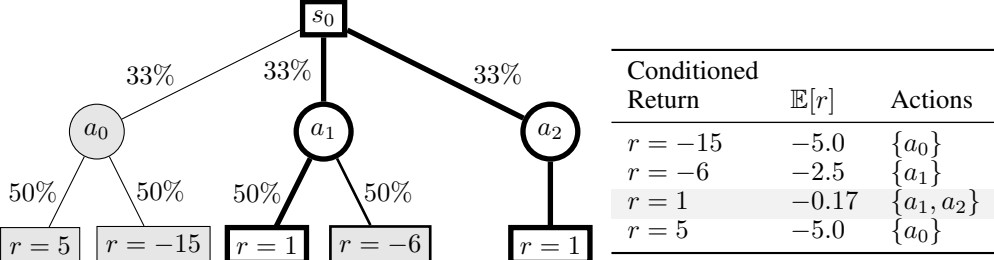

Figure 1: **Left:** A simple gambling environment with three actions where return-conditioned algorithms such as Decision Transformer [1] will fail, *even with infinite data*. The optimal action, $a_2$, will always grant the agent 1 reward while gambling ($a_0$ and $a_1$) give a stochastic amount of reward. The numbers (33%) above each action represent the data collection policy. **Right:** The second column represents the performance of a policy behaviorally cloned from trajectories achieving the reward in the first column. For example, conditioned on the agent receiving $r = 1$, a third of trajectories take $a_1$ and two-thirds take $a_2$. Averaging the returns $-2.5 \times 1/3 + 1 \times 2/3 = -0.17$. **No matter how much data the model is trained on, when conditioned on receiving a reward of** $1$**, it will always gamble and take** $a_1$ **some of the time, rather than just taking** $a_2$**, which guarantees the reward.**

objective can be expressed as minimizing some distance metric $D(\cdot, \cdot)$ between a target goal $z$ and the *information statistics* $I(\tau)$ of the trajectory generated by a conditional policy $\pi(a|s, z)$:

$$\min_{\pi} \mathbb{E}_{z \sim p(z), \tau \sim p_{\pi_z}(\tau)}[D(I(\tau), z)] \tag{2.1}$$

where $p(z)$ is the goal distribution, $p_{\pi_z}(\tau)$ is the trajectory distribution $(s_1, a_1, s_2, a_2, \ldots)$ obtained by running rollouts in an MDP following policy $\pi(a|s, z)$. In this section, we refer to the goal presented to an agent as $z$ and the function that calculates which goal was achieved in a particular trajectory $\tau$ as its statistics $I(\tau)$. As discussed in Furuta et al. [18], this framework includes a number of popular algorithms, including RvS algorithms that condition on target returns (e.g., [1, 2, 8, 6]) or goals (e.g., [7, 19]). For example, in Decision Transformer [1] cumulative discounted reward is used as the trajectory statistics $I(\tau) = \sum_t \gamma^t r_t$ and a reasonable choice for $D(\cdot, \cdot)$ could be the squared error between the target and achieved return.

Furuta et al. [18] justify using the supervised learning approach (RvS) to minimize Equation 2.1 by claiming that trajectories with statistics $I(\tau)$ act as *optimal* examples of how to act to achieve $D(I(\tau), z) = 0$ for an agent whose goal is to achieve $z = I(\tau)$. However, in stochastic environments, the actions taken in these trajectories may not be optimal since **a trajectory may achieve a goal due to uncontrollable randomness in the environment rather than from the agent's own actions.** In fact, as discussed in Paster et al. [19], not only can the actions taken in these "optimal trajectories" actually be sub-optimal, but they can be arbitrarily bad depending on the policy used to collect the data, the environment dynamics, and the choice of distance metric.

In this section, we propose to sidestep this issue by limiting the types of statistics $I(\tau)$ we are interested in to those that are not affected by uncontrollable randomness in the environment. We show first that RvS policies trained to reach such goals, under the assumption of infinite data and model capacity, learn optimal policies for achieving such goals and that goals that are independent from environment stochasticity are the only goals that RvS can reliably achieve. Importantly, we argue that limiting the form of trajectory statistics $I(\tau)$ in this way is not prohibitive and that desirable quantities can be transformed to fit this property in an intuitive way (e.g., return turns into expected return when optimized to be independent of environment stochasticity). Finally, we propose an approach for automatically finding such trajectory statistics using adversarial learning.

## 2.2 Stochasticity Independent Representations

**Definition 2.1** (Consistently Achievable). A goal $z$ is *consistently achievable* from state $s_0$ under policy $\pi(a|s, z)$ if $\mathbb{E}_{\tau \sim p_{\pi_z}(\tau|s_0)}[D(I(\tau), z)] = 0$.

A natural question is under which circumstances will the RvS approach actually minimize Equation 2.1 to zero? Clearly the approach works empirically on deterministic environments [1, 2] since

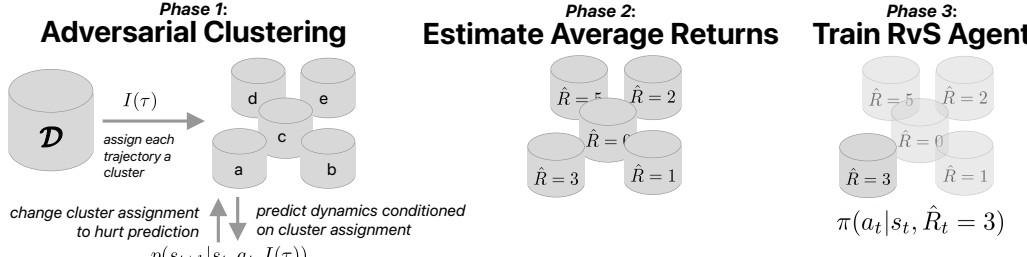

Figure 2: ESPER learns a policy that conditions on a desired *expected* return. In phase 1 of the algorithm, a function is learned using adversarial learning that assigns each trajectory in the dataset to a cluster such that the cluster assignments do not contain information about the stochastic outcomes of the environment that can help a dynamics model cheat to predict next states. In phase 2, the average return in each cluster is calculated. In phase 3, an RvS agent is trained to predict the next action given the current state and the estimated average return.

simply replaying an action sequence that achieved a goal once will achieve it again. While not all statistics of a trajectory will lead to consistently achievable goals under policies trained with RvS, **the supervised learning approach will minimize Equation 2.1 if $I(\tau)$ is independent from environment stochasticity**.

**Theorem 2.1.** Let $\pi_{\mathcal{D}}$ be a data collecting policy used to gather data used to train an RvS policy, and assume this policy is trained such that $\pi(a_t|s_0, a_0, \ldots, s_t, z) = p_{\pi_{\mathcal{D}}}(a_t|s_0, a_0, \ldots, s_t, I(\tau) = z)$. Then, for any goal $z$ such that $p_{\pi_{\mathcal{D}}}(I(\tau) = z|s_0) > 0$, goal $z$ is *consistently achievable* iff $p_{\pi_{\mathcal{D}}}(s_t|s_0, a_0, \ldots, a_{t-1}) = p_{\pi_{\mathcal{D}}}(s_t|s_0, a_0, \ldots, a_{t-1}, I(\tau) = z)$.

**Proof.** [Sketch] The forward direction of the theorem is proven by rewriting $\mathbb{E}_{\tau \sim p_{\pi_z}(\tau|s_0)}[D(I(\tau), z)]$ to be taken over $p_{\pi_z}(\tau|s_0, I(\tau) = z)$ using the independence assumption combined with the form of the policy. For the other direction and the full proof, refer to Appendix A.1. $\qquad\square$

**Remarks.** The above theorem has two major implications. First is that in stochastic environments, many of the most common statistics such as final states or cumulative discounted rewards are often correlated with environment stochasticity and therefore are unlikely to work well with the behavioral cloning approach. For example, in the gambling environment discussed in Figure 1, the achieved reward is correlated with the uncontrollable randomness present when gambling, and therefore when filtering an offline dataset for trajectories that achieve a high reward, the resulting trajectories will have unrealistic environment dynamics because we ignore the unsuccessful attempts (i.e., the agent always wins the money when gambling) and an agent behavioral cloned on these trajectories will not actually end up achieving a high reward consistently.

Theorem 2.1 also gives a solution to ensure asymptotically that RvS policies will consistently achieve goals: use trajectory statistics that are independent from the stochasticity in environment transitions. This requires the statistic function $I(\tau)$ to return the same value for different trajectories where the only difference is in the environment transitions, making $I(\tau)$ essentially act as a cluster assignment where each trajectory within a cluster has the same value of $I(\tau)$ and has environment transitions sampled from the true distribution $p(s_{t+1}|s_t, a_t)$. These clusters can be thought of as datasets each generated by distinct policies in the same environment. For example, in the gambling environment each cluster could include trajectories that all took the same action.

Ultimately, we want to use a target *average return* to control an RvS agent. Luckily, given any $I(\tau)$ that satisfies Theorem 2.1, we can transform it by using the average returns of trajectories within each cluster as the statistic instead. In the next section, we describe a practical algorithm for learning an $I(\tau)$ that satisfies Theorem 2.1, finding the average return of trajectories with the same value of $I(\tau)$ (i.e. in the same "cluster"), and training an RvS policy conditioned on these estimated average returns.

## 2.3 Learning Stochasticity-Independent Representations

In this section we propose a method for learning such trajectory representations, which we call ESPER, short for **e**nvironment-**s**tochasticity-inde**pe**ndent **r**epresentations. Rather than use trajectory return as the statistic $I(\tau)$, which is often dependent on environment stochasticity and therefore may result in unexpected behavior, ESPER uses a neural network to learn statistics that are independent from environment stochasticity. It does so in three phases: first, it uses an auto-encoder framework to learn a discrete representation $I(\tau)$ acting as a cluster assignment for each trajectory along with an adversarial loss that changes the representation in order to hurt the predictions of a dynamics model; second, a model learns to predict average trajectory return from the learned representation; and third, an RvS agent is trained to predict actions taken when conditioned on a state and estimated average return.

For some trajectory $\tau = (s_1, a_1, r_1, s_2, a_2, r_2, \ldots)$, ESPER trains the following parameterized models in addition to a vanilla RvS policy $\pi_\xi$:

$$
\begin{aligned}
\text{Clustering model:} \quad & I(\tau) \sim p_\theta(I(\tau)|\tau) \\
\text{Action predictor:} \quad & a_t \sim p_\theta(a_t|s_t, I(\tau)) \\
\text{Return predictor:} \quad & \hat{R} = f_\psi(I(\tau)) \\
\text{Transition predictor:} \quad & s_{t+1} \sim p_\phi(s_{t+1}|s_0, a_0, \ldots, s_t, a_t, I(\tau))
\end{aligned}
$$

**Adversarial clustering.** Here, we will focus on learning representation in the form of clustering assignments. The goal is to learn a clustering assignment per trajectory $I(\tau)$ to minimize the discrepancy between the true environment transition $p(s_{t+1}|s_t, a_t)$ and the estimated transition $p_\phi(s_{t+1}|s_t, a_t, I(\tau))$ from the replay buffer when conditioned on $I(\tau)$. Such a representation will satisfy Theorem 2.1. In the first phase of training, discrete cluster assignments produced by the clustering model are trained by taking alternating gradient steps with the following two losses to ensure that information present in $I(\tau)$ does not improve the performance of the dynamics model:

$$
L(\theta) = \mathbb{E}_{I(\tau) \sim p_\theta(I(\tau)|\tau)} \Big[ \underbrace{-\beta_{\text{act}} \log p_\theta(a_t|s_t, I(\tau))}_{\text{policy reconstruction}} + \underbrace{\beta_{\text{adv}} \log p_\phi(s_{t+1}|s_0, a_0, \ldots, s_t, a_t, I(\tau))}_{\text{adversarial loss}} \Big]
\tag{2.2}
$$

$$
L(\phi) = \mathbb{E}_{I(\tau) \sim p_\theta(I(\tau)|\tau)} \Big[ \underbrace{-\log p_\phi(s_{t+1}|s_0, a_0, \ldots, s_t, a_t, I(\tau))}_{\text{dynamics prediction}} \Big]
\tag{2.3}
$$

In this step, the dynamics prediction loss is predicting the next state given the current state, action, and cluster assignment $I(\tau)$ while the adversarial loss tries to change $I(\tau)$ in order to hurt this prediction. Since a constant representation minimizes this loss, we use a policy reconstruction loss to encourage $I(\tau)$ to contain information about the policy used to generate the trajectory and encourage the formation of more than one cluster. $\beta_{\text{act}}$ and $\beta_{\text{adv}}$ are hyperparameters to balance the strength of this policy reconstruction loss with the adversarial loss.

**Estimate cluster average returns.** After clustering, we learn to predict discounted returns $R = \sum_t \gamma^t r_t$ for each cluster.[1] The return predictor parameterized by $\psi$ is trained using the following loss:

$$
L(\psi) = \mathbb{E}_{I(\tau) \sim p_\theta(I(\tau)|\tau)} \big[ \|R - f_\psi(I(\tau))\|_2^2 \big].
\tag{2.4}
$$

The above losses are described such that the model produces a single estimated return per trajectory. However, in practice it can be extended to produce a value for each time-step in a trajectory by using suffixes of trajectories $\tau_{i,t} = (a_t, s_t, r_t, \ldots, a_T, s_T, r_T)$ to generate return predictions $\hat{R}_t$ for each time-step.

**Training policy on predicted returns.** After learning expected future returns for each step in a trajectory, we use the dataset of $(s_t, a_t, \hat{R}_t)$ triples to learn a policy $\pi_\xi(a_t|s_t, \hat{R}_t)$ parameterized by

---

[1]Note that in practice, we condition the return predictor not only on the cluster assignment for the trajectory, but also on the first state and action. This improves performance while still satisfying Theorem 2.1.

| Task | Return-Conditioned RvS (DT) | CQL | ESPER (Ours) |
|------|------|------|------|
| Gambling | -0.02 (0.24) | **1.0 (0.0)** | **1.0 (0.0)** |
| Connect Four | 0.8 (0.07) | 0.61 (0.05) | **0.99 (0.03)** |
| 2048 | 0.57 (0.05) | 0.7 (0.09) | **0.81 (0.06)** |

Figure 3: Comparing the maximum performance of RvS with return conditioning, RvS with ESPER conditioning (ours), and CQL [4], a strong value-based offline-RL baseline. The maximum possible expected return for a policy is 1.0 on all tasks. Since all our tasks are stochastic, return conditioning fails to learn an optimal policy while ESPER performs optimally or near optimally. CQL performs well on the gambling task, but cannot achieve the same level of performance as ESPER on the more complicated tasks. Numbers in parenthesis are standard deviations over 3 seeds.

$\xi$ by predicting the action from the current state and estimated expected return by minimizing the following loss just as in prior RvS works [2, 1]:

$$L(\xi) = \mathbb{E}_{s_t, a_t, \hat{R}_t \sim \mathcal{D}} \Big[ - \log \pi_\xi(a_t | s_t, \hat{R}_t) \Big] \tag{2.5}$$

## 2.4 Implementation

The clustering model is implemented using an LSTM [20], using truncated backpropagation through time for training on long trajectories (up to 1000 time-steps in our experiments). The action predictor, return predictor, and transition predictor are all implemented using MLPs. In practice, we found that the dynamics predictor works well even without conditioning on the entire trajectory history. We use a transformer policy as in Decision Transformer [1] (see section 3.2), although we find that using an MLP is sufficient for strong performance in our benchmark tasks (see appendix A.3). Our clustering outputs a discrete value sampled from a categorical distribution and we use gumbel-softmax [21, 22] in order to backpropagate gradients through it. We use normal distributions with unit variances for the transition predictor. Code for our implementation is available at https://sites.google.com/view/esper-paper. See appendix A.7 for pseudocode for the adversarial clustering step of ESPER. See appendix A.5 for more information about the implementation, including specific hyperparameters for our environments as well as best practices.

## 3 Experiments

We designed our experiments to answer the following questions: **(i)** How well does return-conditioned RvS work on stochastic problems where returns are correlated with environment stochasticity? **(ii)** Does ESPER improve performance, both in terms of maximum achievable performance as well as in correlation between the target and achieved returns? **(iii)** Does the degree to which the learned representations in ESPER are independent from environment stochasticity measurably affect the performance of the learned agent?

### 3.1 Stochastic Benchmark Tasks

Prior offline RL methods including Decision Transformer have been tested primarily on deterministic or near-deterministic environments such as continuous control tasks from the D4RL dataset [13] and Atari games [16]. To get a better idea of how these methods and our approach work on realistic, stochastic environments, we created three new benchmark tasks. Since we are evaluating in an offline setting, each task consists of a stochastic environment as well as one or several data collection policies in order to gather the offline dataset.

**Gambling.** An illustrative gambling environment described in Figure 1 with only 3 actions. The offline dataset for this task is 100k steps collected with a random policy.

**Connect Four.** A game of Connect Four, a tile-based game where players compete to get four-in-a-row, against an opponent that does not place a tile in the rightmost column with a probability of 20%. The agent gets 1 reward for winning, 0 for a draw, and $-1$ for losing. The offline dataset for this task

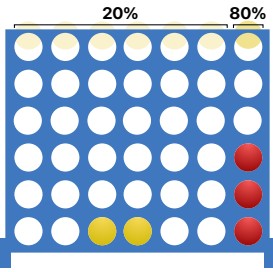 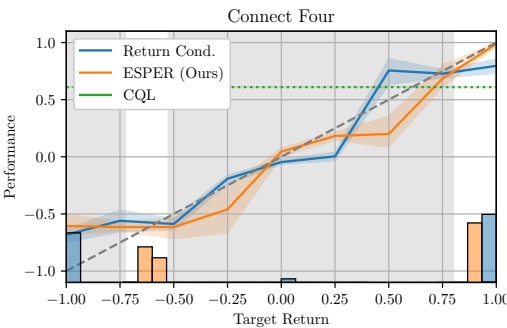

Figure 4: We consider Connect Four with a stochastic opponent that fails to place a piece in the rightmost column with a 20% chance. In this environment, the agent gets a reward of 1 for winning, 0 for a draw, and −1 for losing. The histogram represents the distribution of returns each method is trained on and out-of-distribution regions for ESPER are shaded. On in-distribution returns, ESPER achieves performance (y-axis) closer to the target performance (x-axis) than return conditioning.

is 1M steps collected using a mixture of an $\epsilon$-greedy policy and a policy that always places tiles in the rightmost column.

**2048.** A simplified version of 2048 [23], a sliding puzzle game where alike numbers can be combined to create tiles of larger value, where the agent gets a reward of 1 by creating a 128 tile and gets no reward otherwise. The offline dataset for this task is 5M steps collected using a mixture of a random agent and an expert policy trained using PPO [24].

Detailed information about each environment can be found in Appendix A.4.

### 3.2 Baselines

We compare our method primarily against Decision Transformer [1], which trains a transformer [25] to predict the next action conditioned on a sequence of previous states, actions, and return-to-go targets. Our method uses the same model and training code, except rather than condition on target returns we condition on expected returns learned using ESPER. While we conduct our experiments using transformers due to the popularity of Decision Transformer [1], we note that our observations still hold when using MLP policies as in Emmons et al. [2] while using significantly less compute and getting similar performance (see Appendix A.3). We also compare against a strong value-based approach for offline-RL called Conservative Q-Learning (CQL) [4], which has been found to have strong performance on both discrete and continuous offline RL tasks.

### 3.3 Performance in Stochastic Environments

When comparing the performance of our approach, ESPER, with return-conditioned RvS, we found that **ESPER consistently achieves stronger alignment between target return and average performance** while also achieving a higher maximum level of performance when tuning the target return. As shown in Figure 5a, the return-conditioned baseline cannot achieve the maximum performance of 1 while ESPER learns behaviors corresponding to average performances ranging from −5 to 1. In Connect Four (Figure 4), despite seeing many examples of winning trajectories, an RvS agent cannot achieve more than 0.2 average return (corresponding to a win-rate of 60%) while ESPER can both win and lose with perfect accuracy depending on the target performance. In 2048 (Figure 5b), RvS cannot disentangle trajectories where the reward is high simply from luck from trajectories that took good actions and can only win the game about 60% of the time. ESPER learns many modes of behavior, and can be controlled to win anywhere from 30% to 80% of the time.

Additionally, Figure 6 shows that **this lack of performance for the return-conditioned agent is not due to a lack of data**. In the figure, going from 5% data to 100% (5 million frames) does not affect the performance of RvS while the performance of ESPER gets closer to matching the target return with more data. Finally, we compared the maximum performance of RvS and ESPER with the state-of-the-art value-based offline RL algorithm, Conservative Q-Learning (CQL) [4]. As shown in Figure 3, ESPER and CQL achieve perfect performance on the gambling task while RvS cannot get

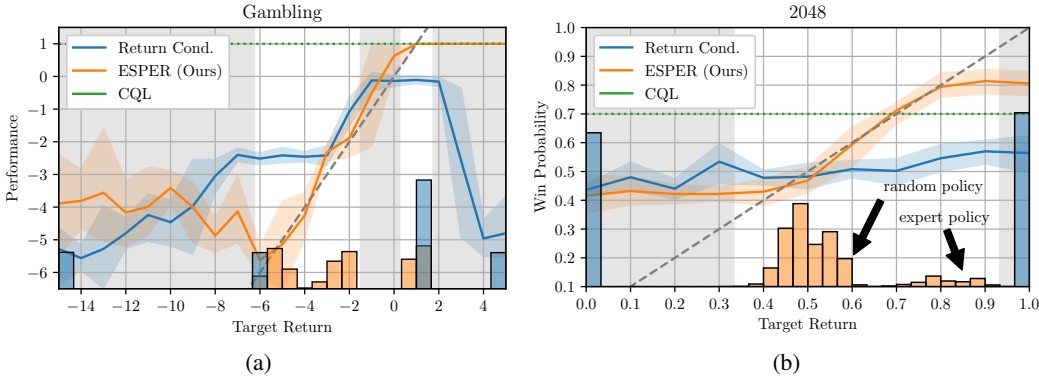

(a)                                                    (b)

Figure 5: The histogram represents the distribution of returns each method is trained on. **Left:** In the illustrative gambling environment, ESPER can achieve a range performances from $-5$ to $1$, while the performance (y-axis) of the return-conditioned agent is not aligned with the target return (x-axis). **Right:** In our modified 2048 task, the agent receives a reward of $1$ for creating a tile of value $128$ and $0$ otherwise. While a return-conditioned agent does not achieve a high level of performance, the ESPER agent disentangles actions from environment stochasticity and is able achieve performance close to the target return. Interestingly, the distribution of expected returns learned by ESPER match the performance-level of the data-collection policies.

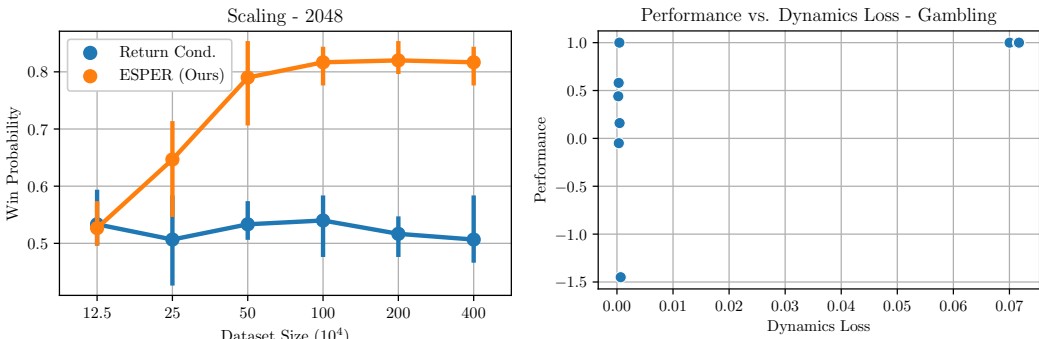

Figure 6: **Left:** While a return-conditioned agent does not improve its performance with more data, ESPER achieves performance (y-axis) closer to the target return (x-axis) when trained on more data. $100\%$ data usage is 5 million frames. **Right:** We trained several agents with different hyperparameters on the gambling task. Notably, agents with trajectory statistics that enabled the dynamics model to "cheat" and get a low loss performed worse than those with representations independent from environment stochasticity, empirically confirming Theorem 2.1.

positive reward. In Connect 4 and 2048, while CQL can get respectable performance (better than the average return of the offline dataset), **ESPER achieves state-of-the-art performance**.

### 3.4 Learned Representations and Behaviors

Rather than simply condition on returns, ESPER conditions on learned expected return values. In addition to plotting the performance against the target return, we also use a histogram to show the distribution of values (returns or learned expected returns) on which the agents are trained. As shown in the histograms (Figure 4, Figure 5a, Figure 5b), ESPER often learns values for many more modes of behaviors corresponding to many different expected returns. Unlike RvS, when an agent is trained on a particular expected return value, the actual average performance of the agent is often close to this value. This is useful for offline settings where tuning the target return in an online environment is not feasible, since **with ESPER one can have confidence in which levels of performance the model will be able to achieve**. In contrast, with return-conditioning, the performance of the return-conditioned policy may not correlate with the returns on which it was trained.

Finally, we empirically measured the relationship between the independence of learned trajectory statistics and the performance of the agent. As shown in Figure 6, we trained several agents with

different hyperparameters on the gambling task and measured the dynamics loss (Equation 2.3) and performance of each agent. Indeed, **agents with trajectory statistics that enable the dynamics model to "cheat" and get a low loss performed worse than those with representations independent from environment stochasticity**, empirically confirming Theorem 2.1.

# 4 Related Work

**Offline Reinforcement Learning.** Offline reinforcement learning is a framework where logged data is used to learn behaviors [3]. In contrast to most online RL methods, offline RL avoids the need for often-expensive online data collection by using fixed datasets of trajectories generated by various policies and stitching together observed behaviors in order to optimize a reward function. The main challenge in the offline setting is that offline RL agents cannot collect more data to reduce their uncertainty and therefore offline RL approaches often have some type of value pessimism or policy constraints to keep the learned policy within the support of the offline dataset. Conservative Q-Learning [4] accomplishes this by learning a conservative lower-bound on the value of the current policy and achieve state-of-the-art performance on most offline-RL benchmarks. Other methods rely on ensembles (e.g., [5]) or policy regularization (e.g., [26–28]). Several standardized benchmarks have emerged for testing offline RL agents, including D4RL [13] and RL Unplugged [14]. In contrast to our work, prior methods for offline RL primarily evaluate on deterministic environments, such as robotics and locomotion tasks as well as Atari tasks, which are near-deterministic [4].

**Reinforcement Learning via Supervised Learning.** At the core of our work is the idea of reducing reinforcement learning to a *prediction problem* and solving it via supervised learning. This idea was first proposed by Schmidhuber [6] and Kumar et al. [8], who proposed to learn behaviors by predicting actions conditioned on a desired outcome such as return. Ghosh et al. [7] proposed to use the same approach to solve goal-conditioned tasks and Paster et al. [19] proposed to predict multi-step action sequences, showing a connection between sequence modeling and optimal open-loop planning. Decision Transformer [1] proposed to use a transformer [25] to condition on trajectory history when predicting actions and tested the approach on offline RL rather than online, achieving results competitive with value-based offline RL approaches. Several followup extensions to Decision Transformer were explored, including using non-return goals [18] and using pretrained transformers [29]. In a recent work, Emmons et al. [2] coined the term RvS (reinforcement learning via supervised learning) to describe such approaches. Notably found that a transformer is not necessary to perform well on most tasks, showing that a two-layer MLP with sufficient regularization can actually outperform transformers at a fraction of the computational cost.

No prior RvS approach has been thoroughly evaluated in stochastic environments. Paster et al. [19] describes a counter-example where an RvS agent in Ghosh et al. [7] doesn't converge in a stochastic environment, and proposes a solution. However, the approach can only be used to plan action sequences rather than reactive plans and therefore won't achieve maximal performance in many stochastic settings. Ortega et al. [30] give a high level view of how sequence modeling can be affected by delusions in various problem settings when not treating actions as interventions. Our contribution is a more precise characterization of the problem within the framework of RvS which relates the choice of goal to environmental stochasticity and directly evokes an efficient algorithm for using RvS in stochastic environments without the need for explicit causal inference or intervention by carefully choosing the goals on which the agents conditions.

**Adversarial Learning.** ESPER uses an adversarial loss to ensure that trajectory statistics are independent from environment stochasticity. Adversarial losses are also widely used in fairness (e.g., [31–33]), where it is used to prevent models from using sensitive attributes, and in generative modeling such as in GANs [34].

# 5 Conclusion

As models are scaled up in pursuit of better performance [10], it is important to acknowledge the limitations that cannot be fixed by more scale and data alone. Our work points out an issue with the increasingly popular Decision Transformer [1] and other RL via supervised learning algorithms when making decisions in stochastic environments. We show why such errors occur and show that these approaches work if and only if the goals that they are trained to achieve are independent from

environment stochasticity. We give a practical algorithm that transforms problematic goals such as trajectory returns into expected returns that satisfy this condition. Finally, we validate our approach by testing it on several new stochastic benchmark tasks, showing empirically that our approach vastly outperforms return-conditioned models in terms of alignment between the target and average achieved return as well as maximum performance. We hope that our work gives insight into how predictive models can be used to make optimal decisions, even in stochastic environments, hopefully paving the way for the creation of more general and useful agents.

**Limitations.** There are several limitations and opportunities for future work. First, our approach relies on learning a dynamics model. In environments where a decently-strong dynamics model cannot be learned, the adversarial clustering in ESPER may fail. Second, ESPER relies on adversarial learning, which similarly to GANs [34], can have learning dynamics that are sensitive to hyperparameter choices (for example learning rate). Additionally, just like in k-means clustering, there are many possible cluster assignments for a given offline dataset. The final performance of ESPER can be affected by variance in the adversarial clustering phase. Finally, while we empirically validate our approach with return goals, we did not explore the use other types of goals where conditioning the model on the mean may be less appropriate, such as visual goals.

**Societal Impact.** We believe that this work will result in a positive societal impact, since our approach can help to avoid unexpected behavior when controlling an agent trained using supervised learning. However, we acknowledge that powerful automated decision making algorithms such as ESPER have the potential to make harmful decisions when trained with under-specified objectives or on biased data.

## Acknowledgements

The authors thank Forest Yang, Harris Chan, Dami Choi, Joyce Yang, Sheng Jia, Nikita Dhawan, Jonah Philion, Michael Zhang, Qizhen Zhang, Lev McKinney, and Yongchao Zhou for insightful discussions and helpful feedback on drafts of the paper. We gratefully acknowledge funding from the Natural Sciences and Engineering Research Council of Canada (NSERC) [JB: 2020-06904], the Canada CIFAR AI Chairs Program, Microsoft Research, the Google Research Scholar Program and the Amazon Research Award. Resources used in preparing this research were provided, in part, by the Province of Ontario, the Government of Canada through CIFAR, and companies sponsoring the Vector Institute for Artificial Intelligence (www.vectorinstitute.ai/partners).

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
