# OpenReview forum: "You Can’t Count on Luck: Why Decision Transformers and RvS Fail in Stochastic Environments"
_NeurIPS.cc/2022/Conference — NeurIPS 2022 Accept_

### Official Review · Reviewer_Hn8H · 2022-07-07

**Rating:** 4
**Confidence:** 4
**Soundness:** 2 fair
**Presentation:** 3 good
**Contribution:** 2 fair

**Summary:**

Offline Decision Transformers trained with supervised learning do not account for the fact that some favorable outcomes are due to stochastic dynamics and not to the quality of the policy.

Therefore, by design, some probability mass may be distributed to sub-optimal actions. The solution proposed is to focus on trajectory statistics that are independent of uncontrollable randomness in the environment. From cluster assignments based on these statistics, useful conditioning quantities such as expected returns can be learnt.

Three predictors are optimized alternatively with two losses:

- On the one hand, a transition predictor is trained to predict the next state conditioned on the previous state, action, and the trajectory cluster assignment.
- On the other hand, the cluster assignment predictor is trained to maximize the loss of the transition predictor while maximizing the likelihood of a policy predictor trained to predict the action conditioned on the current state and cluster assignment.

Finally, another predictor is trained to predict the return conditioned on a cluster assignment. It is then used to label trajectories for policy learning conditioned on expected returns.

The authors propose a benchmark of three stochastic environments and build offline datasets with various policies ranging from random policies to experts. Empirically, their approach outperforms naive return-conditioning and a Conservative Q-Learning baseline.

**Questions:**

On a simple Atari environment considered in the original [Decision Transformer](https://arxiv.org/pdf/2106.01345.pdf) paper (e.g. Pong or Breakout), would it be possible to run your set of experiments with an actual Decision Transformer instead of an MLP? Their [code](https://github.com/kzl/decision-transformer) and dataset of offline trajectories are available.



**Limitations:**

The main limitation of this work is the lack of empirical evidence beyond toy environments and multi-layer perceptrons. While I agree that MLPs can be used instead of Transformers in the MDPs considered, the question of whether Decision Transformers fail in complex stochastic environments is not answered.

**Strengths And Weaknesses:**

# Strengths

- To the best of my knowledge, this method is new. It is simple to understand, well motivated, and easy to implement.
- Claims made at the end of the introduction are supported by experimental results.
- The paper is mostly easy to follow and understand.
- The authors highlight a potential key limitation of Decision Transformers trained with offline trajectories in stochastic environments. As Transformers become more popular as agents, such insights are beneficial to the community.

# Weaknesses

The paper is titled “Why Decision Transformers Fail in Stochastic Environments” but not a single Decision Transformer is trained nor evaluated in this work. While the authors provide evidence that MLP-based behavioral cloning agents conditioned on returns fail in toy stochastic environments, they do not illustrate this phenomenom in more complex environments with recent architectures. For instance, Decision Transformers in [Atari with sticky actions](https://arxiv.org/abs/1709.06009) or [Crafter](https://arxiv.org/abs/2109.06780) could be considered.

---

> ### Author Response · Authors · 2022-08-02
> **Reply to reviewer Hn8H (1/2)**
>
> We would like to thank the reviewer for their review and suggestions.
>
> > The paper is titled “Why Decision Transformers Fail in Stochastic Environments” but not a single Decision Transformer is trained nor evaluated in this work. While the authors provide evidence that MLP-based behavioral cloning agents conditioned on returns fail in toy stochastic environments, they do not illustrate this phenomenon in more complex environments with recent architectures.
>
> We understand your concerns about not using Decision Transformers in our experiments and we include updated results below running our experiments using the Decision Transformer codebase. We would like to justify our original choice, however. In this work, we focus on how the training objective itself leads to a failure of RvS (including Decision Transformer) in stochastic environments. We stress that our discoveries are model-agnostic. Our theory holds regardless of architecture and it should not matter whether the neural network tasked with predicting the next action has access to the current state (RvS with an MLP) or the state-action-return history (Decision Transformer). We chose to include Decision Transformers in the title since Decision Transformers are the most well-known instance of RvS, and we do in-fact show theoretically why Decision Transformers can fail in stochastic environments. However, with recent work showing that empirically (in MDPs) Transformers provide no measurable gain while making experiments far slower [1], we opted to run our experiments with MLPs.
>
> We have redone our experiments to directly build on top of the Decision Transformer codebase to both test the stochastic environments from the paper (show that Decision Transformer fails empirically on stochastic environments) as well as to run ESPER (to show that our approach can fix it, even when using a Transformer). Our updated figures are available here: https://drive.google.com/drive/folders/1vXpYhrjXFYq7Lm4_l2gR58E02VD_UVjV?usp=sharing
>
> As predicted by our theory, Decision Transformer results show the same general trend as the MLP version, failing to act optimally on all three of the benchmark tasks. By replacing the return-to-go conditioning with the learned average returns from ESPER, Decision Transformer achieves a close alignment between the target and achieved performance and learns to perform optimally.
>
> We additionally ran our experiments on the three D4RL Mujoco tasks (deterministic) from the Decision Transformer paper, and found that ESPER performs as well as Decision Transformer in settings without environmental stochasticity.
>
> | **Dataset**   | **Environment** | **Return-to-go Conditioning** | **ESPER (ours)** | **DT**          |
> |---------------|-----------------|-------------------------------|------------------|-----------------|
> | Medium-Expert | hopper          | 3600                          | **89.95±13.91**  | **79.64±34.45** |
> | Medium-Expert | walker          | 5000                          | **106.87±1.26**  | **107.96±0.63** |
> | Medium-Expert | half-cheetah    | 6000                          | **66.95±11.13**  | 42.89±0.35      |
> |               |                 |                               |                  |                 |
> | Medium        | hopper          | 3600                          | 50.57±3.43       | **59.46±4.74**  |
> | Medium        | walker          | 5000                          | **69.78±1.91**   | **69.7±7.12**   |
> | Medium        | half-cheetah    | 6000                          | **42.31±0.08**   | **42.32±0.39**  |
> |               |                 |                               |                  |                 |
> | Medium-Replay | hopper          | 3600                          | **50.20±16.09**  | **61.94±16.99** |
> | Medium-Replay | walker          | 5000                          | **65.48±8.05**   | **63.77±2.82**  |
> | Medium-Replay | half-cheetah    | 6000                          | **35.85±1.97**   | **36.88±0.36**  |

---

> > ### Author Response · Authors · 2022-08-02
> > **Reply to reviewer Hn8H (2/2)**
> >
> > > The main limitation of this work is the lack of empirical evidence beyond toy environments and multi-layer perceptrons. While I agree that MLPs can be used instead of Transformers in the MDPs considered, the question of whether Decision Transformers fail in complex stochastic environments is not answered.
> >
> > We believe that our current experiments provide sufficient empirical evidence to back up our claims that RvS can fail on stochastic environments (ranging from toy tasks to tasks that require complex reasoning about stochasticity). Our work primarily focuses on providing novel insight and theory on the training objectives of RvS (including Decision Transformer). The main contribution is to provide a precise characterization of when such a training objective would likely fail to achieve the desired behavior (when the goal is dependent on environmental stochasticity) and use this theory to devise a method for using RvS in stochastic environments. Therefore, we chose our tasks to directly test the ability of our method to handle stochasticity, especially where return-conditioning fails. Connect Four (vs. a stochastic opponent) and 2048 both require non-trivial reasoning about stochasticity in order to be solved despite having simple observation spaces.
> >
> > We hope that we have answered your concerns about the architecture used (by adding Decision Transformer experiments as well as arguing why our findings are architecture agnostic) and the choice of environments on which we included in our evaluations (we chose tasks that require complex decision making under stochasticity, not necessarily tasks that have complex observation spaces). Given that you acknowledge the strength, novelty, and importance of our contributions, we would appreciate it if you could raise the given score. If you still have questions, we would be happy to answer.
> >
> > [1] "RvS: What is Essential for Offline RL via Supervised Learning?" (Emmons et al. 2021)

---

> > ### Comment · Reviewer_Hn8H · 2022-08-07
> > **Response to rebuttal**
> >
> > Thank you for your reply and for running these additional experiments. I understand it is quite difficult to write rebuttals and provide new results in such a short timeframe.
> >
> > *“As predicted by our theory, Decision Transformer results show the same general trend as the MLP version, failing to act optimally on all three of the benchmark tasks. By replacing the return-to-go conditioning with the learned average returns from ESPER, Decision Transformer achieves a close alignment between the target and achieved performance and learns to perform optimally.”*
> >
> > On Connect Four, the trend is not the same as the vanilla Decision Transformer shows a target/performance alignment very similar to ESPER, even though ESPER edges supervised learning for the maximum target return.
> >
> > Besides, the performance of the supervised learning approach substantially increased (0.24 → 0.8) when moving from MLPs to Transformers. This illustrates why results with MLPs in toy environments are not a good proxy for what would happen with other architectures in more complex environments.
> >
> > Overall, my initial concerns still remain. The experiments do not go beyond very basic environments to empirically validate ESPER and it remains unclear whether Decision Transformers actually fail in stochastic environments of interest (e.g. Atari with sticky actions or Crafter).

---

> > > ### Author Response · Authors · 2022-08-08
> > > **Justification for our choice of environments (1/2)**
> > >
> > > Thanks for your response and for reiterating your concerns about the empirical evaluation. We would like to take another stab at convincing you that (1) our criteria for choosing benchmark tasks is sensible given the core contribution of the paper; (2) Atari and Crafter are not appropriate benchmarks under these criteria; and (3) there is precedent for the tasks that we chose from prior work on RL in stochastic environments.
> > >
> > > **On the Improved Connect Four Results with Decision Transformer.** The opponent in our Connect Four benchmark is very close to optimal and any suboptimal action can quickly lead to a loss, so an increase in the quality of behavioral cloning (by using a more powerful architecture, for example) will understandably increase the performance of both ESPER (which is already perfect) and return-conditioning. This does not change that a return-conditioned agent fails to perform optimally under the presence of stochasticity in this environment because even with perfect behavioral cloning, the objective is still incorrect. If anything, the difference in performance by changing architectures shows that as the environment gets more complex, the harmful effect of this misaligned objective can be overshadowed by imperfections in the other steps of the process such as the behavioral cloning itself. This does not make the wrong objective any less of a problem. This is one reason why we choose to test our hypotheses and proposed algorithm in environments with simple observation and action spaces where it is easier to isolate the effect of the objective and stochasticity.
> > >
> > > **Regarding our criteria for choosing benchmark tasks.** The primary objective of our paper is to provide insight, both empirically and theoretically, on why Decision Transformers and RvS in general fail to perform optimally in stochastic environments. This led us to choose benchmark tasks with minimal confounding factors (partial observability, high-dimensional observation and action spaces, etc.) to show that even in the absence of these other challenges, Decision Transformer cannot solve even simple stochastic tasks like our gambling environment. In fact, we view our results as being a *more* significant contribution than if we had only shown that Decision Transformers fail in the most “complex” stochastic environments, since researchers and practitioners should be especially concerned if their offline-RL algorithm cannot solve even simple tasks. As we scaled up the complexity of our experiments, we scaled the complexity of stochasticity and decision making rather than observation complexity since this is of primary concern to us.
> > >
> > > **Atari and Crafter are not Appropriate Benchmarks.** Sticky actions in Atari ensure that optimal action sequences cannot simply be replayed in order to get good performance. However, agents that don’t take stochasticity into account already perform well on Atari with sticky actions, suggesting that the stochasticity introduced is not enough to make it a good benchmark for testing decision making in stochastic environments. For example, MuZero Unplugged [1], which outperforms both Decision Transformer (which disables sticky actions during evaluation) and CQL on the Atari 1% dataset, achieves this performance while planning with a fully deterministic model. As shown in Table 7 of their paper [1], there is little difference in performance with sticky actions enabled or disabled, even in online RL.  (In fact, mean performance gets slightly better when sticky actions are enabled.) Crafter, on the other hand, is a relatively new benchmark task *for online RL* meant to test “generalization, wide and deep exploration, representation learning, and long-term reasoning and credit assignment” [2]. While this may be an interesting candidate for an offline-RL task, testing these capabilities of RvS and Decision Transformer are out of the scope of this paper. We instead opt to evaluate our algorithm on tasks where reasoning under stochasticity is the primary challenge.

---

> > > > ### Author Response · Authors · 2022-08-08
> > > > **Justification for our choice of environments (2/2)**
> > > >
> > > > **Precedent for our chosen tasks.** We stress that our choice of benchmark tasks (an illustrative toy task, Connect Four, and 2048) are in line with prior work in evaluating RL on stochastic environments. For example, in an extension to MuZero that introduces a stochastic model and planning [3] (ICLR 2022 Spotlight), the authors do not evaluate performance on Atari but rather on two stochastic environments: Backgammon and 2048, which provides a challenge for the deterministic version of MuZero and for 2048,  “despite its simplicity, model-free approaches have traditionally struggled to achieve high performance” [3]. Additionally, they show that their changes do not hurt the performance of MuZero in the deterministic game of Go. While unintentional, our experiments ended up following the precedent that this paper establishes: we evaluated on a simple board game (Connect Four), 2048, and on Mujoco tasks to ensure that our changes do not hurt performance.
> > > >
> > > > **Summary.** We believe that our choice of experiments for our empirical evaluation are justified and sufficient to back up the claims in our paper. Our results, with return-conditioned RvS agents like Decision Transformer failing to learn optimal policies even in simple environments, should cause researchers and practitioners to think carefully about the limitations of these methods, and we believe the theory and algorithm presented in our paper can serve as a starting point for addressing them. Finally, our choice of environments is well in-line with prior, well-regarded work on extending RL algorithms to support stochastic environments [3].
> > > >
> > > > Given this, we sincerely hope that you will reconsider your position and raise your score to an accept for a paper that you described as simple, well-motivated, well-supported by experimental results, easy to understand, and beneficial to the community.
> > > >
> > > > [1] “Online and Offline Reinforcement Learning by Planning with a Learned Model” (Schrittwieser et al. 2021)
> > > >
> > > > [2] “Benchmarking the Spectrum of Agent Capabilities” (Hafner 2022)
> > > >
> > > > [3] “Planning in Stochastic Environments with a Learned Model” (Antonoglou et al. 2022)

---

> > > > ### Comment · Reviewer_mkK7 · 2022-08-08
> > > > **On The Choice Of Environments**
> > > >
> > > > I agree with the authors that Crafter is not an appropriate benchmark to request here. It is not a widely adopted benchmark for this area, does not isolate the fundamental problem being addressed and is large enough in scale to be its own publication.
> > > >
> > > > I agree with reviewer Hn8H that experiments in Atari with sticky actions would strengthen the contribution, but with any empirical paper we can always ask for more experiments. Given the precedent set by Antonoglou et al. [ICLR 2022] at last year's conference, why raise the bar on expected environments now? If DT already fails on these simple stochastic environments, why does the reviewer expect it might still work in more complex stochastic environments?

---

### Official Review · Reviewer_bsi4 · 2022-07-10

**Rating:** 7
**Confidence:** 4
**Soundness:** 3 good
**Presentation:** 3 good
**Contribution:** 3 good

**Summary:**

This paper first identify that the existing offline "reinforcement learning via supervised learning" (RvS) methods that conditioned on returns/outcomes (e.g. Decision Transformers) can only reliably achieve outcomes that are independent from environment stochasticity. In other words, they do not consider whether the offline demonstration results are achieved by pure luck.

Based on this observation, the authors proposed learning learn environment-stochasticity-independent representations of trajectories, by adversarially clustering offline experiences by data collection policy, to disentangle actions from environment stochasticity. Trajectories in each cluster can be annotated with the average return for trajectories. These pieces together give ESPER, environment-stochasticity-independent representation. RvS with ESPER achieves substantial improvements on several simple stochastic environment benchmarks: Gambling (an illustrative single-step stochastic environment), Connect Four with a stochastic opponent, and 2048.

**Questions:**

As mentioned in the above section, I wonder if we can have more discussion around potential limitations and future challenges of the proposed approach.

nitpicks
* In the Figure 2 caption, it's unclear what "... such that the dynamics within each cluster matches the environments dynamics ..." means. I suggest rephrasing it.
* I recommend spelling out "ESPER" at least in introduce. It doesn't appear until Sec 2.3, which makes it difficult to follow.


**Limitations:**

The potential negative societal impact is addressed. However, the limitation is not sufficiently discussed, which I think is also my main concern about his paper and discussed in the "Strengths And Weaknesses" section.

**Strengths And Weaknesses:**

Strengths:
* I appreciate the insightful observation that existing RvS can only reliably achieve outcomes that are independent from environment stochasticity. This paper presents a clear illustration to explain why this could be an issue. I believe future RvS research could potentially benefit a lot from this work.
* The proposed solution is theoretically sound. Empirically, it shows decent improvements, compared to return conditioning, on several simple benchmarks, including Gambling, Connect Four, and 2048. I think these are enough proof that the issue does exist, and that the proposed method ESPER achieve some initial successes.

Weaknesses:
My concerns are about whether this paper has discussed enough limitations of the proposed solution (i.e. clustering by policy) and enough future challenges. Besides high-dimensional observation space, I think clustering by behavior/policy implies that, when the data is collected with many behavioral modes, the number of clusters need to be able to scale with that. For that reason, I think we will perhaps see that it struggles with high-dimensional and continuous action space. Potentially, (state-dependent) no-ops and near-no-ops could also lead to unnecessary complexity. Furthermore, my haunch is that clustering by behavior will go against behavioral compositionality, making it difficult to interpolate desired outcomes, as opposed to return-conditioning, and thus probably go against the authors' desire to "pave the way for creation of more general agent" as stated in the conclusion section.

Given the early position of this paper (which focus on a limited set of empirical evaluations), I would like to see much more discussion about the potential limitations and future challenges of the proposed approach, to help future works to extend this contribution. If these could be sufficiently addressed or clarified during rebuttal, which I think would not be very difficult, I will increase my rating to 6 or 7 to recommend acceptance.

---

> ### Author Response · Authors · 2022-08-02
> **Reply to reviewer bsi4 (1/2)**
>
> We would like to thank the reviewer for their thoughtful review.
>
> > My concerns are about whether this paper has discussed enough limitations of the proposed solution (i.e. clustering by policy) and enough future challenges.
>
> We agree that our paper exposes an issue with RvS in stochastic environments, provides a precise characterization of the types of goals RvS will work with, and uses this theory to propose an initial solution to the problem to show one way that it can be solved. We will revise our limitations section to better discuss the weaknesses of our approach. In our opinion, the primary weaknesses are:
>
> - The approach relies on learning a dynamics model, so in environments where a decently strong dynamics model cannot be learned, ESPER won’t work.
> - ESPER relies on adversarial learning, which similarly to GANs, can have learning dynamics that are sensitive to hyperparameter choices (for example learning rate).
> - Just like in k-means clustering, there are many possible cluster assignments for a given offline dataset. The final performance of our algorithm can be affected by variance in the clustering.
>
> > Besides high-dimensional observation space, I think clustering by behavior/policy implies that, when the data is collected with many behavioral modes, the number of clusters need to be able to scale with that. For that reason, I think we will perhaps see that it struggles with high-dimensional and continuous action space.
>
> We agree that ESPER should learn many clusters when running on data with many behavioral modes. However, in our experiments we already use ~1 million clusters on 2048 and Connect Four (4 categorical variables of dimension 32), so we have reason to believe that ESPER can work well in practice even with a very large number of clusters. Additionally, we are including new results, running ESPER on top of Decision Transformer on three D4RL Mujoco tasks with continuous actions. Updated figures running ESPER with a Decision Transformer (building on top of their codebase) are available here: https://drive.google.com/drive/folders/1vXpYhrjXFYq7Lm4_l2gR58E02VD_UVjV?usp=sharing.
>
> Results on Mujoco tasks:
>
> | **Dataset**   | **Environment** | **Return-to-go Conditioning** | **ESPER (ours)** | **DT**          |
> |---------------|-----------------|-------------------------------|------------------|-----------------|
> | Medium-Expert | hopper          | 3600                          | **89.95±13.91**  | **79.64±34.45** |
> | Medium-Expert | walker          | 5000                          | **106.87±1.26**  | **107.96±0.63** |
> | Medium-Expert | half-cheetah    | 6000                          | **66.95±11.13**  | 42.89±0.35      |
> |               |                 |                               |                  |                 |
> | Medium        | hopper          | 3600                          | 50.57±3.43       | **59.46±4.74**  |
> | Medium        | walker          | 5000                          | **69.78±1.91**   | **69.7±7.12**   |
> | Medium        | half-cheetah    | 6000                          | **42.31±0.08**   | **42.32±0.39**  |
> |               |                 |                               |                  |                 |
> | Medium-Replay | hopper          | 3600                          | **50.20±16.09**  | **61.94±16.99** |
> | Medium-Replay | walker          | 5000                          | **65.48±8.05**   | **63.77±2.82**  |
> | Medium-Replay | half-cheetah    | 6000                          | **35.85±1.97**   | **36.88±0.36**  |
>
> ESPER matches the performance of Decision Transformer on D4RL Mujoco tasks, showing both that using the learned returns from ESPER does not hurt the performance of RvS compared to using return-conditioning even in (the tested) deterministic environments, and that the approach can scale to continuous actions.

---

> > ### Author Response · Authors · 2022-08-02
> > **Reply to reviewer bsi4 (2/2)**
> >
> > > Furthermore, my haunch is that clustering by behavior will go against behavioral compositionality, making it difficult to interpolate desired outcomes, as opposed to return-conditioning
> >
> > We believe that clustering by behavior is no less expressive in terms of behavioral compositionality and ability to interpolate desired outcomes than return-conditioning for both theoretical and empirical reasons. Theoretically, return-conditioning is simply a special case of conditioning with clustering (like in ESPER) with a large number of clusters, where each trajectory and return can be in its own cluster. In fact, in a deterministic environment, the optimal clustering for ESPER is to put each trajectory in its own cluster. Therefore, we view conditioning based on clusters as being a strict generalization of return-conditioning in the sense that in addition to considering each trajectory individually, the algorithm can group trajectories together to better measure performance under stochasticity. We will update the paper to make this more clear. Empirically, ESPER can match the performance of DT on Mujoco tasks, showing that in practice this clustering does not hurt performance.
> >
> > > If these could be sufficiently addressed or clarified during rebuttal, which I think would not be very difficult, I will increase my rating to 6 or 7 to recommend acceptance.
> >
> > If you believe that we have sufficiently addressed your concerns about the limitations by adding to the discussion of limitations as well as answering your concerns about whether the clustering approach can scale to complicated environments and whether it will hurt performance compared to return-conditioning, we would appreciate an updated score. If you have any additional questions, we would be happy to answer them.

---

> > > ### Comment · Reviewer_bsi4 · 2022-08-06
> > > **How do you interpolate between discrete clusters that are unrelated to each other?**
> > >
> > > It is true that return-conditioning may be considered as a special case of clustering. However, in that case, behaviors are mapped to a 1-d space, where each point correspond to a return value. However, in the case of ESPER, the class id does not seem to have any semantic meaning. Can you elaborate on how it is possible to interpolate between discrete clusters that don't have any meaningful labels? If it is possible to do so, please show a concrete example using what is described in this paper. If that's indeed a limitation, I think there should be some easy fix to this and I do not think that's necessary for this paper, and instead, the only request I have is to document it.

---

> > > > ### Author Response · Authors · 2022-08-07
> > > > **Interpolation with ESPER**
> > > >
> > > > Thanks for the question. My understanding of your question is the following - please let me know whether I’m correct. The concern is that even though ESPER may generalize return-conditioning, the clusters learned by ESPER are represented in a space with no semantic meaning. This may hurt the interpolation performance compared to when returns are used as labels.
> > > >
> > > > To clarify, in ESPER, first, clusters are learned and then each cluster is “relabeled” with the average return within the cluster (“phase 2” of the algorithm). The raw cluster assignments are not fed as input into the policy but rather the average returns of trajectories within that cluster, giving the labels semantic meaning. With this understanding, the ability to interpolate between behaviors is no more of a problem in ESPER than in return-conditioned RvS since the inputs to the policy will be the same. Let us know whether this helps to clarify. We will add this to the paper to clarify how ESPER generalizes return-conditioned RvS.

---

> > > > > ### Comment · Reviewer_bsi4 · 2022-08-08
> > > > > **Thanks for the clarification**
> > > > >
> > > > > I see. That was my misreading. I don't have further questions and concerns. Also thanks for updating the manuscript. I will update my rating.

---

> > > > ### Author Response · Authors · 2022-08-08
> > > > **New version of the paper**
> > > >
> > > > We have uploaded a new version of the paper that contains an updated limitations section, updated figures and experiments, as well as a section to briefly explain how ESPER generalizes return-conditioning. Note that we have temporarily moved the limitations and societal impact section to the appendix for the rebuttal phase - we will move them back to the main text in the final version when we are allowed an additional page. Please let us know whether these changes and our clarifications merit the increase in score that you mentioned in your original review. If not, we are happy to make additional changes to the text and answer any other questions.
> > > >
> > > > Additionally, we would appreciate it if you could weigh in on the discussion with reviewer Hn8H, who is concerned about our choice of environments.

---

### Official Review · Reviewer_mkK7 · 2022-07-11

**Rating:** 7
**Confidence:** 4
**Soundness:** 3 good
**Presentation:** 3 good
**Contribution:** 3 good

**Summary:**

This paper proposes an approach to handling the challenge of applying RL via supervision methods to stochastic environments. The method is empirically demonstrated to outperform two recent baselines across three environments.

**Questions:**

1) Please clarify how the observation regarding the fundamental problem of applying RL via supervision agents to stochastic environments differs from or compliments the findings of "Shaking the foundations: delusions in sequential models for interaction and control" Ortega et al. [Arxiv 2020]. This could improve my rating of the significance of contribution (1).

2) Why was CQL not included in Figures 4-6? Does including it provide further insight into the relative strengths and weaknesses of these approaches? This could improve my rating of the significance of contribution (2).

3) On lines 243-244, the paper notes "ESPER can both win and lose with perfect accuracy depending on the target performance" but misses an opportunity to discuss why it doesn't do well at drawing (0 target return). Is this because this return is not included in the training distribution due to the policies used to generate the offline dataset of trajectories? This question would help clarify my current understanding of the empirical results.

**Limitations:**

The authors have adequately addressed the limitations and potential negative societal impact of their work. I would be very interested in seeing the follow up study proposed regarding testing the performance of ESPER on other types of goals as I agree this may require further innovation.

**Strengths And Weaknesses:**

At the bottom of page 2, the paper claims three contributions:
1) the observation that RL via supervision agents "can only reliably achieve outcomes that are independent from environment stochasticity."
2) the proposed method ESPER
3) the stochastic offline RL benchmark tasks used to evaluate ESPER

However:
- observation (1) was previously made in "Shaking the foundations: delusions in sequential models for interaction and control" Ortega et al. [Arxiv 2020]
- the environments (3) contribution cannot be fully evaluated as code is only available for the simplest of the three environments. Open sourcing all environments would improve the significance of this contribution.

ESPER (the proposed method) remains a novel contribution that significantly outperforms two recent related baselines. The empirical evaluation of this method could be improved by including CQL in figures 4-6 and quantifying the correlation between performance and target return currently visualized in Figure 6. I agree with the conclusion the authors reach from this plot, but in this case believe the raw value would be more informative than the plots currently used. The space saved could then be used to include the more insightful results currently in Appendix A.3 on the representations learnt.

---

> ### Author Response · Authors · 2022-08-02
> **Reply to reviewer mkK7 (1/2)**
>
> We would like to thank the reviewer for their insightful comments and suggestions.
>
> > Open sourcing all environments would improve the significance of this contribution.
>
> We are working to open source our code and we are committed to releasing the code upon publication.
>
> > The empirical evaluation of this method could be improved by including CQL in figures 4-6 and quantifying the correlation between performance and target return currently visualized in Figure 6.
>
> We agree that indicating the level of performance our method reaches versus CQL is important. RL via supervised learning learns a policy that is conditioned on a target return, so figures 4/5 are meant to show the alignment between target and achieved performance. We did not include CQL in figures 4-6 since CQL only learns one maximum-performance policy. In an updated plot, we will add horizontal lines in figures 4/5 to indicate the level of performance that CQL achieves (currently this information is available in the table in Figure 3). We will also redo figure 6 in a future version to plot performance vs. dataset size in a more compact way and bring the plot from Appendix A.3 to the main text, which we agree is an insightful result.
>
> > Please clarify how the observation regarding the fundamental problem of applying RL via supervision agents to stochastic environments differs from or compliments the findings of "Shaking the foundations: delusions in sequential models for interaction and control" Ortega et al. [Arxiv 2020]. This could improve my rating of the significance of contribution (1).
>
> We acknowledge the connection with this paper and we will add to the related work section to highlight the similarities and differences. Specifically, our work focuses on the RvS scenario, where the policy is learned offline with no interventions. This goal-conditioned scenario is mentioned briefly in Ortega et al., which says that “goal-conditioned planning does not work in the general case [because] the goal is an action, and conditioning on it will reveal information about the latent task parameter, causing a delusion” (Remark 6). In section 4.2, Ortega et al. discuss how offline adaptation and control is currently an open problem and difficult due to the presence of confounding factors.
>
> ESPER is an algorithm that solves precisely this problem in the RvS case by characterizing the types of goals which will lead to delusions. Rather than try to solve a hard or impossible causal inference problem, we change the problem to one that is easily solved by changing the goal we condition on. While Ortega et al. gives a high level view of how sequence modeling can be affected by delusions in various problem settings when not treating actions as interventions, our contribution is a more precise characterization of the problem within the framework of RvS which relates the choice of goal to environmental stochasticity and directly evokes an efficient algorithm for using RvS in stochastic environments without the need for explicit causal inference or intervention. We believe that the contributions of our paper justify an increase in your rating of the significance of our contribution. If we haven’t convinced you, we would be happy to answer any additional questions.
>
> > Why was CQL not included in Figures 4-6? Does including it provide further insight into the relative strengths and weaknesses of these approaches? This could improve my rating of the significance of contribution (2).
>
> Since this information is available in the table in figure 3, we don’t believe that adding it to the plots will provide any further insight into the relative strengths and weaknesses of the approaches. Plots 4-6 show the alignment between targeted and achieved performance, so we can’t directly compare to CQL (which only learns one policy) in this context. We will update the plots to include CQL as a horizontal line showing that CQL outperforms return-conditioned RvS, while ESPER outperforms both.

---

> > ### Author Response · Authors · 2022-08-02
> > **Reply to reviewer mkK7 (2/2)**
> >
> > > On lines 243-244, the paper notes "ESPER can both win and lose with perfect accuracy depending on the target performance" but misses an opportunity to discuss why it doesn't do well at drawing (0 target return). Is this because this return is not included in the training distribution due to the policies used to generate the offline dataset of trajectories? This question would help clarify my current understanding of the empirical results.
> >
> > We believe that since trajectories that end in a draw are rare in the training distribution, this mode of behavior was not found in the clustering phase of ESPER. We will add a discussion about why certain modes of behavior may not be found during the clustering phase of ESPER. Similar to k-means clustering, there are many cluster assignments for a given offline dataset. RvS benefits from clusters which capture many modes of behavior in the data (behaviors with both weak and strong performance, for instance). The final performance of our algorithm may be affected by the variance in the clusters discovered. In practice, we use a large number of clusters (~1M) and we find that this lets the clustering algorithm discover diverse behaviors. We suspect that in the Connect Four experiment, since drawing behavior only represents a small number of samples, that the clustering phase didn’t assign it to a cluster. That said, when we re-ran the experiments for the rebuttal using the Decision Transformer codebase, we found that it is now able to draw in Connect Four.
> >
> > Updated figures for new experiments that we ran on top of the Decision Transformer codebase can be found here: https://drive.google.com/drive/folders/1vXpYhrjXFYq7Lm4_l2gR58E02VD_UVjV?usp=sharing. We additionally ran experiments on the (deterministic) D4RL Mujoco tasks and found that ESPER performs no worse than Decision Transformer (using learned average returns does not hurt performance in cases where return-conditioning already works):
> >
> > | **Dataset**   | **Environment** | **Return-to-go Conditioning** | **ESPER (ours)** | **DT**          |
> > |---------------|-----------------|-------------------------------|------------------|-----------------|
> > | Medium-Expert | hopper          | 3600                          | **89.95±13.91**  | **79.64±34.45** |
> > | Medium-Expert | walker          | 5000                          | **106.87±1.26**  | **107.96±0.63** |
> > | Medium-Expert | half-cheetah    | 6000                          | **66.95±11.13**  | 42.89±0.35      |
> > |               |                 |                               |                  |                 |
> > | Medium        | hopper          | 3600                          | 50.57±3.43       | **59.46±4.74**  |
> > | Medium        | walker          | 5000                          | **69.78±1.91**   | **69.7±7.12**   |
> > | Medium        | half-cheetah    | 6000                          | **42.31±0.08**   | **42.32±0.39**  |
> > |               |                 |                               |                  |                 |
> > | Medium-Replay | hopper          | 3600                          | **50.20±16.09**  | **61.94±16.99** |
> > | Medium-Replay | walker          | 5000                          | **65.48±8.05**   | **63.77±2.82**  |
> > | Medium-Replay | half-cheetah    | 6000                          | **35.85±1.97**   | **36.88±0.36**  |

---

> > ### Comment · Reviewer_mkK7 · 2022-08-04
> > **Reply to authors**
> >
> > Thank you for your detailed replies to all reviewers. Having read them all, I remain in support of the paper as it contributes novel insight into the fundamental problem of RvS in stochastic environments. In particular, this response improved my understanding of this paper's contribution compared to prior work [Ortega et al. 2020].
> >
> > I also appreciate the proposed revisions to Figures 4-6 and additional experiments run in comparison to DT. Assuming updated pdfs can still be submitted during the discussion period, it would be reassuring to see these changes implemented in the paper before updating my score.
> >
> > Finally, given that the paper contributes to understanding the broad class of RvS algorithms I would encourage the authors to consider revising the title to "You can't count on luck: Why **RvS** fails in stochastic environments" instead of the implementation specific DT name currently used.

---

> > > ### Author Response · Authors · 2022-08-08
> > > **New version of the paper**
> > >
> > > We have uploaded a new version of the paper that contains the updated experiments and figures, as well as the updated related works section. Note that we have temporarily moved the limitations and societal impact section to the appendix for the rebuttal phase - we will move them back to the main text in the final version when we are allowed an additional page. We have also changed the title to reflect that our paper applies not only to Decision Transformer but also to other RvS methods.
> > >
> > > We hope that our changes and clarifications can contribute to a stronger accept. Additionally, we would appreciate it if you could weigh in on the discussion with reviewer Hn8H, who is concerned about our choice of environments.

---

> > > > ### Comment · Reviewer_mkK7 · 2022-08-08
> > > > **RE: New version of the paper**
> > > >
> > > > Thank you for promptly sharing an updated pdf. I have updated the rating in my original review to reflect my increased opinion of the contribution made by this paper.

---

### Meta-Review · Area_Chair_GzLw · 2022-08-25

**Recommendation:** Accept
**Confidence:** Certain

**Metareview:**

The authors explore a fundamental limitation of Decision Transformers and related RL via supervised learning approaches when applied to stochastic environments. They propose a new and simple approach that clusters experiences to disentangle action quality from environment stochasticity. Their ESPER approach achieves large improvements on a number of simple, stochastic environments including Gambling, Connect Four and 2048.

The reviewers were all satisfied by the novelty, technical soundness and relevance of this work for the NeurIPS community, but were initially of mixed opinions about the selection of challenge domains. Having discussed this point at length with the reviewers, I am satisfied with the authors' choice of environments for two reasons: (1) they allow the specific shortcomings of previous methods to be isolated and addressed directly, and (2) they are consistent with the environments used by previous related work published in top-tier conferences. I am recommending this paper for acceptance accordingly.

**Award:**

No

---

### Decision · Program_Chairs · 2022-09-14

Accept